# Improving Knowledge of Audit and Feedback among Health Care Professionals in Sicily

**DOI:** 10.3390/healthcare11141987

**Published:** 2023-07-09

**Authors:** Augusto Ielo, Maria Cristina De Cola, Francesco Corallo, Giangaetano D’Aleo, Agata Mento, Davide Cardile, Irene Cappadona, Maria Pagano, Placido Bramanti, Rosella Ciurleo

**Affiliations:** 1IRCCS Centro Neurolesi Bonino-Pulejo, 98124 Messina, Italy; augusto.ielo@irccsme.it (A.I.); francesco.corallo@irccsme.it (F.C.); giangaetano.daleo@irccsme.it (G.D.); davide.cardile@irccsme.it (D.C.); irene.cappadona@irccsme.it (I.C.); maria.pagano@irccsme.it (M.P.); rossella.ciurleo@irccsme.it (R.C.); 2Papardo Hospital, 98158 Messina, Italy; agatamento@gmail.com; 3Faculty of Psychology, Università Degli Studi eCampus, Via Isimbardi 10, 22060 Novedrate, Italy

**Keywords:** audit, feedback, training, healthcare professionals, survey

## Abstract

Audit and Feedback (A&F) is considered one of the most significant tools for implementing continuous Quality Improvement (QI) in the healthcare field. The audit process is a structured inspection of professional practice against known standards or targets. The results of this inspection are subsequently feedback from professionals in order to implement an improvement process. The Italian Ministry of Health has recently funded the network project EASY-NET, with the main objective of evaluating the effectiveness of A&F strategies to improve healthcare practice and equity in various clinical and organizational settings in seven Italian regions. The Sicily region is represented within the EASY-NET project by the IRCCS Centro Neurolesi Bonino-Pulejo of Messina as the Work Package 7 (WP7). One of the objectives of the WP7 is to assess mechanisms and tools to enhance the effectiveness of A&F strategies. The purpose of this study is to investigate the influence that training interventions can have on improving knowledge of A&F strategies among healthcare professionals. The study employed a quasi-experimental design with a pretest-posttest evaluation strategy. The participants’ initial knowledge of A&F strategies was evaluated through a baseline survey. Subsequently, the participants attended an online training workshop led by A&F experts, and a follow-up survey consisting of the same set of questions was conducted at the end of the process. Results showed statistically significant positive changes in the level of knowledge of A&F among participants following the training intervention. Furthermore, dividing the participants into two subgroups based on their professional background revealed significant differences in the level of knowledge of A&F methodologies between the observed categories of healthcare professionals. In conclusion, the study revealed that training interventions can be facilitators to implementing effective A&F programs.

## 1. Introduction

The process of translating clinical guidelines into practice is one of the most challenging yet potentially valuable factors of evidence-based medicine [1]. Quality Improvement (QI) is a systematic, formal application of specific methodologies to analyze the practice performance and efforts to improve it. It is vital that healthcare professionals know how to use QI methods in their work and in making major practice enhancements [2]. Tools used to implement QI include the distribution of printed educational material to healthcare professionals [3,4], the planning of educational meetings or workshops [5], and Audit and Feedback (A&F) strategies [6].

The term A&F describes a set of heterogeneous strategies widely used for continuous QI in healthcare. Although A&F can be implemented in different ways and with different goals, it is essentially based on two interdependent stages: The Audit stage is defined as “a summary of clinical performance over a specific period of time”; the Feedback consists in the provision of that summary (feedback) to individual practitioners, teams, or health care organizations [7]. According to the most recent Cochrane reviews A&F interventions generally lead to small but potentially important improvements in professional practice [6]. Furthermore, recent studies suggest that A&F interventions may have a high potential to be cost-effective [8]. Nevertheless, it is still unclear whether A&F is more effective than other healthcare QI strategies, especially when combined with one of these strategies [9].

The Italian network project EASY-NET [10], funded by the Italian Ministry of Health, has the main objective of evaluating the effectiveness of Audit and Feedback strategies to improve healthcare practice and equity in various clinical and organizational settings. Seven Work Packages (WPs), each located in a different Italian region, are participating in the project. The IRCCS Centro Neurolesi Bonino-Pulejo of Messina, Sicily is responsible for the WP7 and recently described a new study protocol with the aim of assessing the effectiveness of A&F strategies in both Acute Myocardial Infarction (AMI) and ischemic stroke setting, from acute to rehabilitation process of care [11].

Feedback, contextual and recipient variables can influence whether and how health workers respond to A&F, and thus they can impact the success of A&F implementation as part of routine practice [12]. Among the recipient’s variables, such as individual abilities and beliefs, even more relevant, at least in an initial stage, are their knowledge about A&F strategies and the dissemination of change culture aimed at improving clinical practice.

To date, in Italy, A&F strategies are still not widely used and known, even among healthcare professionals in hospital settings.

As part of the EASY-NET project, one of the objectives of the WP7 is to understand recipient variables as the knowledge of mechanisms and tools to enhance the effectiveness of A&F strategies. This paper provides insights on designing an effective training model that aims to measure and improve knowledge of A&F methodologies among healthcare workers, also taking into consideration their different levels of specialization or professional paths. The proposed training model was designed to be eventually reiterated over time, with a view to continuous quality improvement.

## 2. Materials and Methods

### 2.1. Study Design and Setting

A quasi-experimental design with pretest-posttest evaluation was used to conduct the study. The participants attended an online training workshop led by A&F experts which was tailored based on the knowledge gaps identified from a baseline survey. The main objective of the online training workshop was to transfer knowledge related to planning, conducting and analyzing A&F interventions, and implementation of a multidimensional system of quality indicators integrated into corporate governance processes. The participants underwent a survey pre and post-intervention. In the reporting phase, all the results were pulled together and summarized with a view to the continuous improvement of the process. Figure 1 describes the flow of the study.

The study was carried out in the IRCCS Centro Neurolesi Bonino-Pulejo of Messina, Sicily. The subjects who were involved in the study are healthcare workers in the context of cardio and cerebrovascular emergency and rehabilitation from Sicilian hospitals located in the Messina, Catania and Palermo areas and specialized in the IMA e ictus treatment. The online training workshop recipients were also involved in the EASY-NET project.

### 2.2. Inclusion/Excusion Criteria

All the healthcare professionals who registered for the workshop after its publicization were eligible to participate in the study. Subjects who did not submit either the baseline or follow-up surveys according to the established time schedules were excluded from the study.

### 2.3. Study Population

Thirty-eight healthcare professionals attended the workshop, and therefore, were eligible to participate in this study. Of these, only twenty-six participants (mean ± SD age: 47.9 ± 9.1 years; 46.2% male) were enrolled in the study as they submitted both the baseline and follow-up surveys. Participants were then assigned to two groups (PG1: n = 13, PG2: n = 13) based on their level of specialization or professional background. A detailed description of the study population is reported in Table 1.

### 2.4. Training Interventions

A panel of experts who are part of the EASY-NET project designed the training intervention. The panel consisted of three professionals, respectively, specialized in (i) risk management, (ii) quality improvement strategies and clinical audit, and (iii) statistics and quality indicators. The online workshop has been subdivided into three separate interventions: the first covered the main aspects of risk management, quality improvement and A&F in the healthcare sector; the second consisted of an in-depth look at clinical audit; the final part covered the data collection methodologies and the assessment of quality indicators in clinical audit. The contents of the workshop were tailored according to the participants’ initial knowledge of A&F which was assessed from the results of the ad-hoc questionnaire described in the next section.

### 2.5. Assessment

An ad-hoc survey has been developed by the same panel of experts who designed the training interventions in order to assess the participants’ initial (T0) and final (T1) knowledge of A&F. This survey included three topics or question group (QGs) each consisting of five questions: general aspects of A&F (QG1); clinical A&F (QG2); quality indicators and data collection in A&F (QG3), for a total of 15 questions (Q1 to Q15). The detailed list of the questions included in the survey is shown in Table 2.

In order to analyze differences in outcomes between different types of healthcare professionals, the pool of participants was further divided into two subgroups based on their level of specialization or professional background: the first participant group (PG1) consists of healthcare providers as physicians and psychologists while the second participant group (PG2) consists of nurses, physical therapists and neurophysiopathology technicians. Following the T1 survey, a satisfaction questionnaire was conducted to obtain the feedback of participants. The purpose of the satisfaction survey was to analyze the experience of the participants in order to highlight any areas that could be improved in the training interventions.

### 2.6. Data Collection

The data were collected employing an online survey administration tool (Google Forms). Several studies have supported data collection with the use of online surveys, especially in settings where paper-and-pencil methods cannot be carried out due to time and space constraints [13]. The baseline and follow-up surveys were prepared and links to the Google Forms were circulated via email to all the subjects registered for the workshop. The baseline survey was sent to participants after their enrollment. The follow-up questionnaire was sent two days after the training intervention. At the end of the process, the collected data were exported to a spreadsheet format directly from the Google Forms tool for further cleaning and analyses.

### 2.7. Statistical Analysis

Data analysis was performed using R version 4.2.2, considering a *p* < 0.05 as statistically significant. Due to the small size of the sample non-parametric analysis was performed. The Wilcoxon signed-rank test was used to compare scores between baseline and follow-up; the same test was used to compare the scores of each subgroup between baseline and follow-up (intra-group analysis), while a Mann–Whitney U-test was used for inter-group analysis. The Chi-squared test was used to compare proportions.

An analysis of covariance (ANCOVA) was performed to evaluate whether the means of the outcome scores at follow-up (dependent variable) are equal across the participants’ academic and professional background (categorical independent variable) while statistically controlling for the effects of the scores at baseline (covariate). The model had the test score at T1 as the dependent variable, the categorical variable ‘GROUP’ (1 = PG1; 0 = PG2) as the independent variable, and the test score at T0 as a covariate. Assumptions of homogeneity of regression slopes and homogeneity of variance were assessed by ANOVA and Levene’s test, respectively.

## 3. Results

### 3.1. Intra-Group Analysis

An analysis of the results at baseline showed that participants scored a lower percentage of correct answers (less than 50%) on three questions. Participants gave five correct answers to Q5 (two for PG1 and three for PG2), nine correct answers to Q14 (five for PG1 and four for PG2) and eight correct answers to Q15 (six for PG1 and two for PG2).

After the training workshop, significant improvement in the median total score was observed: from 9 to 13 (*p* < 0.001). Statistically significant differences (*p* < 0.001) were also found by analyzing the results of each QG separately. As per the QGs, overall higher scores at both T0 and T1 were observed for the QG2 set of questions. Figure 2 shows the differences between scores at T0 and T1 for each of the QGs.

Furthermore, an analysis of the scores for each individual question at T0 and T1 was performed. An increase in the percentage of correct answers for all questions at follow-up was observed, although statistically significant differences found were limited to Q1 (χ^2^ (1) = 4.15, *p* = 0.042), Q5 (χ^2^ (1) = 13.08, *p* < 0.001), Q7 (χ^2^ (1) = 4.71, *p* = 0.030), Q10 (χ^2^ (1) = 11.53, *p* = 0.001), Q13 (χ^2^ (1) = 6.00, *p* = 0.014), Q14 (χ^2^ (1) = 6.27, *p* = 0.012) and Q15 (χ^2^ (1) = 13.32, *p* < 0.001).

### 3.2. Sub-Groups Analysis

No significant differences were found in demographic characteristics between the two groups (PG1 and PG2) as described in Table 1. Inter-group analysis showed no statistical significance between the total scores both at baseline (*p* = 0.221) and follow-up (*p* = 0.473). On the other hand, the intra-group analysis revealed significant differences in both PG1 and PG2 groups (*p* = 0.006 and *p* = 0.003, respectively). The intra-group analysis was also performed for each QG as described in Table 3.

Figure 3 shows the differences between scores at T0 and T1 for each of the QGs, for each of the two groups of participants.

Further analysis of the scores, as the percentage of correct answers per participant group at T0, the percentage of correct answers for all questions at T0 and T1, and the number of correct answers for all questions at T0 and T1 per participant group are available in the Appendix A.

The data met the assumptions of homogeneity of regression slopes and homogeneity of variance. The interaction term was not considered in the ANCOVA model fitting since ANOVA has shown that this term does not contribute significantly to the covariate models. As described in Table 4 and Figure 4, the different level of specialization or professional background has significantly influenced the improvement of outcomes (t = 3.241; *p* = 0.004).

### 3.3. Satisfaction Survey

The satisfaction questionnaire was anonymously submitted by 38 workshop participants. The level of satisfaction was rated on a scale from 1 to 5, where 1 signifies very low and 5 signifies very high. The relevance of the topics covered, relative to the individual continuing education needs was rated very high by 22 participants (57.9%) and high by 16 participants (42.1%). The educational quality of the program was rated very high by 22 participants (57.9%), high by 15 (39.5%) and medium by 1 participant (2.6%). Lastly, the usefulness of the event for training/updating was rated very high by 23 participants (60.5%) and high by 15 participants (39.5%).

## 4. Discussion

Significant differences between the baseline and the follow-up scores might reflect the improved knowledge of A&F strategies among the participants. In fact, the results of our study show that involving professionals may be the best option to improve clinical processes.

As suggested by the literature, high-quality processes have been shown to improve the outcomes of care in various settings. Ventura et al. [14] demonstrated that acute myocardial infarction (AMI) patients exposed to optimal quality of care along the multicomponent continuum of care have better 1-year survival. Recent reviews of the literature show that the implementation of QI programs leads to improvements in stroke rehabilitation care. Notably, Shafei et al. [15] reported that in more than 90% of cases, improvements were noted through the implementation of multicomponent interventions including A&F strategies, whereas Cappadona et al. [16] evaluated seven studies stating that A&F appears to be effective in improving the quality of care for stroke patients in the rehabilitation phase.

On the whole, the sustained adherence to clinical practice guidelines and the use of quality improvement initiatives enables physicians to optimize patient outcomes through the practice of safer, evidence-based medicine [17]. Indeed, A&F programs can increase clinicians’ adherence to guideline recommendations [18], and guideline recommendations are often used as the basis for clinical audits [4].

QI requires a systematic process involving all hospital staff with a focus on A&F strategies. Having a clinical audit team and department within every healthcare organization has long been considered an important factor in facilitating risk reduction, although there are some constraints to overcome [19]. The barriers to conducting the A&F interventions may involve the lack of resources, expertise in all professions which handle the audit programs or advice in project design and analysis, problems between groups or between members of the same group, lack of an overall plan for audit, organizational impediments [20], excessive workload and time constraint, and availability of good data from routine hospital records and administrative health system database [21]. Another important barrier to A&F interventions, which is often underestimated, is the lack of knowledge and unawareness of healthcare professionals about A&F and its advantages. Training and education programs on A&F strategies aimed at practitioners receiving such interventions could promote the effectiveness of A&F programs.

From the literature searches performed, no recent articles dealing with rising awareness of A&F methodologies were found. We believe that the one discussed in this study is a relevant issue and that our study is innovative in this regard.

This study, although carried out in a small context, aims at providing interesting insights on the level of knowledge of A&F strategies among healthcare professionals, and on the impact that training interventions can have in raising awareness of these methodologies. Indeed, low knowledge of A&F strategies among clinical practitioners receiving such interventions might limit their effectiveness.

At baseline, the low percentage of correct answers to question Q5 reveals that before the training interventions, most of the participants were unaware of the benefits that an audit process can bring to the people involved. Moreover, other major knowledge gaps at baseline concerned the topic of quality indicators and data collection in A&F.

The study demonstrated an overall statistically significant positive change in the level of knowledge of A&F strategies among healthcare professionals following the training intervention. As per the topics covered in the survey, overall higher scores were obtained in answer to the questions related to the Clinical Audit subject.

Dividing the participants into two groups revealed significant differences in the level of knowledge of A&F methodologies between the observed categories, which may depend on the level of specialization or professional background of healthcare providers participating in the surveys, as suggested by the intra-group analysis. In fact, the PG1 consisting of physicians and psychologists performed better than PG2 both at baseline and follow-up. The absence of significant results in the inter-group analysis is indicative of the improvement in outcomes in both groups of professionals at baseline and follow-up. The PG2 consisting of nurses, physical therapists and neuro-physiopathology technicians showed a wider knowledge gap in the QGs related to general aspects of A&F (QG1) and quality indicators and data collection in A&F (QG3).

The COVID-19 pandemic has strongly boosted the development of online training. Although this form of distance learning does not prove very effective because the recipients can be poorly motivated and fail to maintain a high attention span, the improvement in the median total score of the survey suggested that the online program described was found to be quite effective. It is likely that the issue of A&F, though little known, has induced in healthcare professionals some awareness of the need to identify a method aimed at improvements in their clinical practice.

Furthermore, the positive feedback from the participants received through the satisfaction survey, showed that issues included in the training interventions have been considered informative and beneficial for their professional growth.

## 5. Future Research

In view of the assumptions above, future actions could include the design of distinct training interventions, tailored for each category of healthcare providers. Furthermore, other types of intervention in addition to the workshop could be carried out. For example, an experimental laboratory for developing ideas among the involved healthcare workers could be developed.

In future studies, the PDCA (Plan–Do–Check–Act) cycle could be readapted into an educational cycle. The process described in this study could be implemented with a view to continuous quality improvement by periodically repeating the design and conduction of the level of knowledge assessment, training interventions and satisfaction surveys.

More tools could be used to evaluate the effect of the training interventions. For example, the Kirkpatrick model is widely used to evaluate the educational effect [22,23].

Finally, future studies with greater sample sizes will be needed in order to evaluate the accuracy of these considerations.

## 6. Conclusions

In the context of the EASY-NET project, the A&F intervention conducted on trained practitioners compared with the A&F intervention conducted on untrained practitioners will allow us to evaluate whether training interventions can be facilitators to implementing effective A&F programs.

## Figures and Tables

**Figure 1 healthcare-11-01987-f001:**
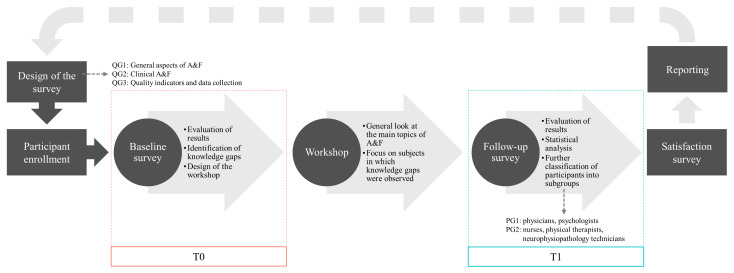
Flow of the study. Legend: QG1–3 = Question Group 1–3; PG1–2 = Participant Group 1–2.

**Figure 2 healthcare-11-01987-f002:**
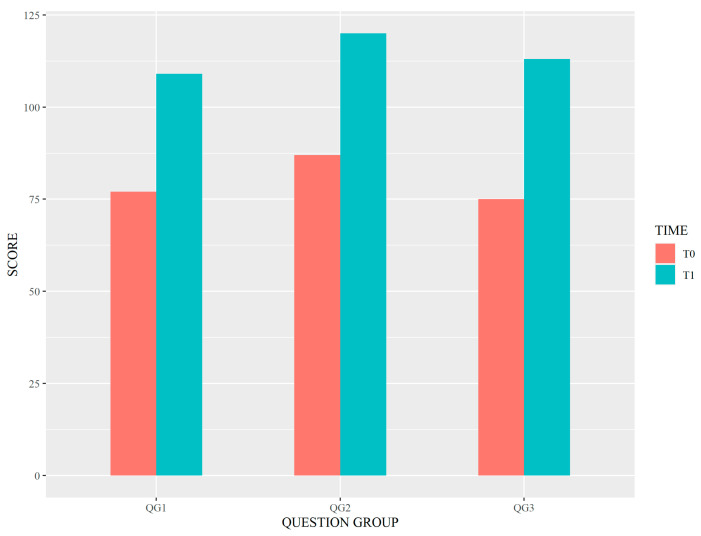
Differences between scores at T0 and T1 for each of the QGs. Legend: QG1–3 = Question Group 1–3.

**Figure 3 healthcare-11-01987-f003:**
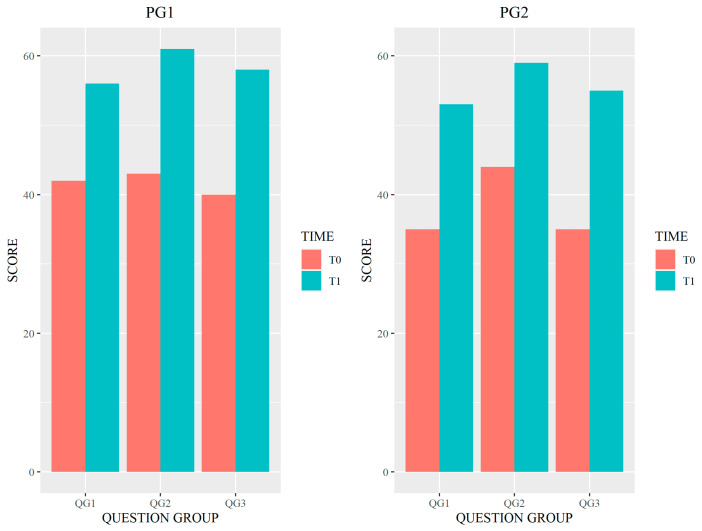
Differences between scores at T0 and T1 for each of the QGs by participant group. Legend: QG1–3 = Question Group 1–3.

**Figure 4 healthcare-11-01987-f004:**
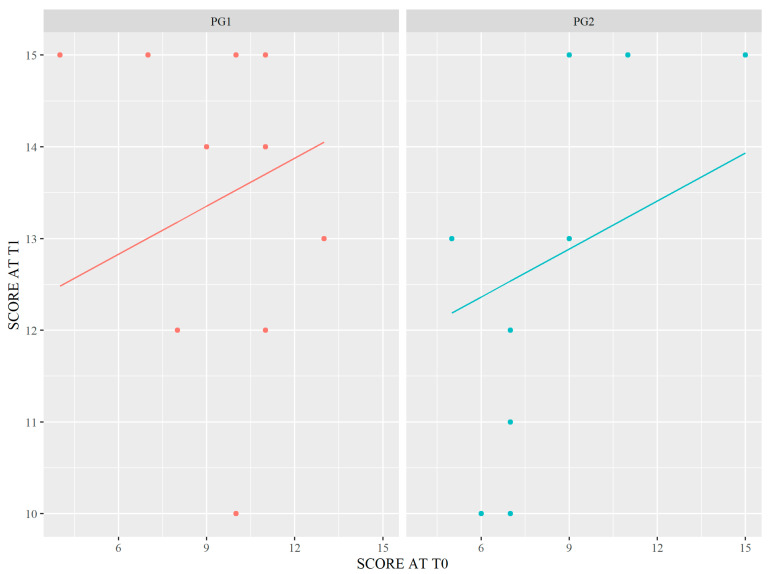
Graphical representation of ANCOVA. Legend: PG1–2 = Participant Group 1–2.

**Table 1 healthcare-11-01987-t001:** Description of the study population.

	All	PG1	PG2	*p*-Value
Participants	26	13 (50.0)	13 (50.0)	-
Male	12 (46.2)	5 (38.5)	7 (53.9)	0.83
Age (years)	47.9 ± 9.1	48.3 ± 11.9	47.5 ± 5.4	0.70

Legend: (PG1) physicians, psychologists; (PG2) nurses, physical therapists, neurophysiopathology technicians. Continuous variables were expressed as mean ± standard deviation, whereas categorical variables as frequencies (percentages).

**Table 2 healthcare-11-01987-t002:** Survey questions.

Question Group	Question Number	Question
QG1	Q1	In which area has the audit process been used extensively in recent times?
	Q2	What is the definition of “internal audit”?
	Q3	What is the definition of “external audit”?
	Q4	What are the characteristics that all types of audit have in common?
	Q5	What is the benefit that an audit process brings to the people involved?
QG2	Q6	What is the definition of “clinical audit”?
	Q7	On what elements is the clinical audit based?
	Q8	What, in order, are the stages of a clinical audit process?
	Q9	How should a clinical audit working group be composed?
	Q10	What is the goal of a clinical audit process?
QG3	Q11	In the context of a clinical audit process, what is an indicator?
	Q12	Which types of studies can be employed for data collection?
	Q13	Which of the following methods should be excluded for data collection?
	Q14	What is the definition of “current data”?
	Q15	Which of the following is not a type of quality indicator?

Legend: (QG) Question Group; (Q) Question.

**Table 3 healthcare-11-01987-t003:** Statistical comparisons of survey scores between baseline (T0) and follow-up (T1), for both groups.

	PG1		*p*-Value	PG2		*p*-Value
	T0	T1		T0	T1	
S.QG1	3.0 (3.0–4.0)	5.0 (4.0–5.0)	**0.013**	2.0 (1.0–4.0)	4.0 (3.0–5.0)	**0.005**
S.QG2	3.0 (3.0–4.0)	5.0 (4.0–5.0)	**0.013**	3.0 (3.0–4.0)	5.0 (4.0–5.0)	**0.008**
S.QG3	4.0 (3.0–4.0)	5.0 (4.0–5.0)	**0.021**	2.0 (2.0–3.0)	5.0 (4.0–5.0)	**0.005**
S.TOT	10.0 (8.0–11.0)	14.0 (12.0–15.0)	**0.006**	7.0 (7.0–9.0)	13.0 (11.0–15.0)	**0.003**

Legend: Scores are in median (first-third quartile); Significant differences are in bold. S.QG1 = Question group 1 scores; S.QG2 = Question group 2 scores; S.QG3 = Question group 3 scores; S.TOT = Survey total scores.

**Table 4 healthcare-11-01987-t004:** Summary table for the ANCOVA.

	Group Coefficient				Adjusted R^2^
	Estimate	Std. Error	t Value	*p* Value	
S.TOT	0.497	0.153	3.241	**0.004**	0.312

Legend: Significant differences are in bold. S.TOT = Survey total scores.

## Data Availability

Datasets are available to download on request. Requests should be directed to the corresponding author: Maria Cristina De Cola, mariacristina.decola@irccsme.it.

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
