# Peer review of "Improving Knowledge of Audit and Feedback among Health Care Professionals in Sicily"

_healthcare, 2023, doi:10.3390/healthcare11141987_

Round 1

Reviewer 1 Report

Dear Authors, Your work is valuable and has practical application. Unfortunately, the study was very small and it was difficult to find a group (26 people who added some in the overall follow-up to the pre- and post-survey results) to be sufficient. The study was also carried out based on very simple tools that are available in teaching (short online pre- and post-surveys). Making a repair also requires some important things: Why were selected professional groups grouped in ten ways, that doctors and psychologists were included, and the second group included nurses, technicians, physiotherapists, and neuropathologists? There is a mention in the construction that this was done based on experience? Is it about work experience? There is no information that psychologists or doctors have the greatest professional experience in this field.   The survey was entered by the same expert team that later conducted the training. Didn't this affect the risk of suggesting a direct answer in the post-training test to get a higher value? Introducing an independent training team and another to check the survey to get the objective value of knowledge.   The survey was conducted on-line/ When do I conduct it before the training? How long after the results of the training did the participants receive links to the research? How to secure, by participation only once filled out the survey and by the answer obtained only by people who took part in the training?   As a result of the findings, it was found that professional preparation influenced the achievement of the results. A PG1 group that has a higher both start and end value than PG2. Is the knowledge creation of the PG1 group proportionate or can the knowledge (common to the delivered values) of the PG2 group be used? The presented drawings are too large, which is visible and they do not look aesthetically pleasing. Out of the cited 21 items of literature, 11 items are from the last 5 years.

Author Response

Dear Reviewer,

we are grateful for your insightful comments on our paper. Please find our answers below.

"Your work is valuable and has practical application. Unfortunately, the study was very small and it was difficult to find a group (26 people who added some in the overall follow-up to the pre- and post-survey results) to be sufficient. The study was also carried out based on very simple tools that are available in teaching (short online pre- and post-surveys)."

Author: We agree with your observations. As we stated in the "Future research" section of our manuscript, further studies will implement more assessment tools and a larger sample.

"Why were selected professional groups grouped in ten ways, that doctors and psychologists were included, and the second group included nurses, technicians, physiotherapists, and neuropathologists? There is a mention in the construction that this was done based on experience? Is it about work experience? There is no information that psychologists or doctors have the greatest professional experience in this field."

Author: Thank you for your observations. To answer your questions, physician and psychologist are both in the management class and have more responsibilities than the other evaluated professionals. In addition, physician and psychologist have more specialized educational backgrounds than the professionals placed in the second group. Our choice comes from these assumptions.

"The survey was entered by the same expert team that later conducted the training. Didn't this affect the risk of suggesting a direct answer in the post-training test to get a higher value? Introducing an independent training team and another to check the survey to get the objective value of knowledge."

Author: We agree with your point. There could be a chance for a bias. However, the training intervention was tailored on the answers given in the baseline survey. So in the case of our study, it was natural that the same team of experts would be responsible for both the creation of the survey and the design of the training interventions. The goal of the EASY-NET project is to give knowledge, so training is provided to fill any gaps on the topic.

"The survey was conducted on-line/ When do I conduct it before the training? How long after the results of the training did the participants receive links to the research?"

Author: Thank you for your observation. Actually, the timeline was not properly described in the paper. To answer your question, the baseline survey was sent to participants after their enrollment. The follow-up survey was sent two days after the training intervention. This timeline has now been added to the revised manuscript.

"How to secure, by participation only once filled out the survey and by the answer obtained only by people who took part in the training?"

Author: The link was sent only to people who decided to register for the event, before and after the training interventions. A unique identifier was assigned to each participant to ensure pre-post matching.

"As a result of the findings, it was found that professional preparation influenced the achievement of the results. A PG1 group that has a higher both start and end value than PG2. Is the knowledge creation of the PG1 group proportionate or can the knowledge (common to the delivered values) of the PG2 group be used?"

Author: Thank you for your question. The knowledge creation of the PG1 group is not proportionate. In fact, we found a greater increase in the score in the second group as shown in Table 3.

"The presented drawings are too large, which is visible and they do not look aesthetically pleasing."

Author: Thank you for the feedback. All the charts have been resized to be more readable. Furthermore, the bars in Figure 2 have been reduced in width to be more aesthetically pleasing.

"Out of the cited 21 items of literature, 11 items are from the last 5 years."

Author: Thank you for your feedback. More recent reference items have been added to the study.

Reviewer 2 Report

This research complements the strategy launched by the Italian Ministry of Health in the EASY-NET network project, in which it aims to improve care practice and equity in diverse clinical and organizational settings in seven Italian regions. The region of Sicily is represented within this project.

PROPOSALS FOR IMPROVEMENT: Considering the abstract and the introduction it is quite clear the proposal. Even so, at the end of the introduction I would add some ideas to attract the reader even more to the importance of this research.

In the discussion it would be advisable to make some more comments, showing that so far what has been published in the scientific literature is not as relevant as this article or has some biases or is incomplete or deals with other issues.

The introduction and conclusions sections should be expanded. It is also recommended to make more recent bibliographical references. 

Author Response

Dear Reviewer,

we are grateful for your insightful comments on our paper and proposal of improvement. Please find our answers below.

"Considering the abstract and the introduction it is quite clear the proposal. Even so, at the end of the introduction I would add some ideas to attract the reader even more to the importance of this research."

Author: Thank you for your observation. Some more notes were added at the end of the introduction to attract the reader even more to the importance of this research.

"In the discussion it would be advisable to make some more comments, showing that so far what has been published in the scientific literature is not as relevant as this article or has some biases or is incomplete or deals with other issues. The introduction and conclusions sections should be expanded."

Author: Some more comments in the discussion were made, so the opening and a final parts were expanded.

"It is also recommended to make more recent bibliographical references."

Author: Thank you for your feedback. More recent reference items have been added to the study.

Reviewer 3 Report

Audit and feedback is a widely used quality strategy that can contribute to improving healthcare quality and patient outcomes. In an audit and feedback process, an individual's professional practice or performance is measured and then compared to professional standards or targets. The results of this comparison are then fed back to the individual. I have no problem with the general remarks, but I would like additional comments on the evaluation of education. I would like you to show a way to turn the PDCA cycle of education. For example, the Kirkpatrick Model is a widely recognized method of evaluating the results of training and learning programs.

Author Response

Dear Reviewer,

we are grateful for your insightful comments on our paper. Please find our answers below.

"I would like additional comments on the evaluation of education. I would like you to show a way to turn the PDCA cycle of education. For example, the Kirkpatrick Model is a widely recognized method of evaluating the results of training and learning programs."

Author: Thank you for your feedback. Your suggestions were added in the “Future research” section of the manuscript.

Round 2

Reviewer 1 Report

I accept the changes made